# Hydrogen Sensor Based on NTC Thermistor with Pt-Loaded WO_3_/SiO_2_ Coating

**DOI:** 10.3390/mi13122219

**Published:** 2022-12-14

**Authors:** Changwei Sun, Ben Xu, Ping Li

**Affiliations:** 1School of Electronic and Electrical Engineering, Bengbu University, Bengbu 233030, China; 2College of Optical and Electronic Technology, China Jiliang University, Hangzhou 310018, China

**Keywords:** hydrogen sensor, NTC thermistor, Pt-loaded WO_3_/SiO_2_

## Abstract

A novel hydrogen sensor based on a negative temperature coefficient (NTC) thermistor with Pt-loaded WO_3_/SiO_2_ coating is proposed and demonstrated experimentally. When the Pt-loaded WO_3_/SiO_2_ film is exposed to the mixture of air and H_2_, the exothermic reactions caused by hydrogen and WO_3_ with the cooperation of the Pt catalyst raise the local temperature of the NTC thermistor and lower its resistance. Hence, hydrogen concentration can be measured by monitoring the voltage across the NTC thermistor in a series circuit. The proposed device has a rapid response time, high sensitivity, and excellent repeatability to hydrogen as well as immunity to humidity, a compact size, a low manufacturing cost, and is easy to use.

## 1. Introduction

Clean energy is attracting a large number of researchers, due to the greenhouse effect, energy shortages, and environmental pollution. Hydrogen has been widely used as a clean, highly efficient, and sustainable energy source in power engineering, chemical industries, the electrical industry, aerospace, and other fields [1,2,3]. However, due to its low ignition point, hydrogen is extremely explosive in ambient air at concentrations ranging from 4% to 75%. Thus, the rapid detection of hydrogen leakage with high sensitivity is extremely important for the safe use of hydrogen. Various types of hydrogen sensors have been developed thus far. The traditional handmade catalytic combustion gas sensor is not influenced by humidity changes, but these sensors have some drawbacks, including difficulty in pairing, poor consistency, high power consumption, and non-interchangeability, and they are vulnerable to poisoning because they have molecules containing siloxane and sulfur [4,5,6]. Semiconductor-type sensors have good stability, selectivity, and sensitivity, but they only work well at high temperatures [7], necessitating the use of an additional high-temperature thermostatic control. Thin-film hydrogen sensors is another important area of research. These thin films are typically made of hydrogen-sensitive materials such as Pd, WO_3_, and other semiconductor oxide or two-dimensional materials, such as graphene, decorated with noble metal nanoparticles, and are coated on the surfaces of devices [3,8,9,10,11]. For example, after absorbing hydrogen molecules, the volume of Pd film expands. This feature can be used to create strain-sensitive optical fiber hydrogen sensors [12] or micro-optomechanical resonator-based hydrogen sensors [13], but the embrittlement effect is still a pressing issue for the stability and sensitivity of Pd-based hydrogen sensors [3,14,15,16]. WO_3_ is another candidate for hydrogen-sensing thin-film owing to its optical properties, such as the gasochromic effect and the refractive index effect [17,18,19,20]. However, when compared to other gases, such as hydrogen sulphide and acetylene, the chemical interaction between WO_3_ and hydrogen is insufficient [3]. As a result, bulk WO_3_ film has no selective sensitivity to hydrogen. To address this issue, doping metal catalysts (such as Pt, Pd, Ag, and others) is a common method of lowering the reaction activation energy [21,22]. When Pt is used as the catalyst, for example, WO_3_ can significantly react with hydrogen at room temperature and continuously generate heat. Because of the exothermic reaction between WO_3_ and hydrogen, some temperature-sensitive optical devices are used for hydrogen sensing by coating a WO_3_ film with a suitable catalyst [23,24,25,26,27]. Furthermore, when the sensors are separated from hydrogen, the intermediate WO_3−x_ is oxidized and reforms WO_3_, allowing the sensors to be reused. These optical hydrogen sensors have high sensitivity and inherent safety, but the sensing system used to detect subtle changes in optical properties, such as wavelength shift [24,25,26,27], is complicated and costly. Fortunately, thermistors offer a more cost-effective and efficient method of precisely measuring temperature variations caused by hydrogen; they are small in size, fast in response, have high measurement accuracy, and are inexpensive [28]. Two-dimensional materials decorated with noble metal nanoparticles, such as graphene decorated with Pd nanoparticles [9,10,11], are a promising combination for hydrogen sensing material due to the large surface area with respect to volume. The advantages of such sensors are their quick response and high sensitivity.

In this paper, a novel hydrogen sensor based on a negative temperature coefficient (NTC) thermistor with Pt-loaded WO_3_/SiO_2_ coating is fabricated and tested. Due to the exothermic reaction of WO_3_ and hydrogen, the temperature of the NTC thermistor rises, and its resistance changes accordingly. Hydrogen concentration can be easily measured by monitoring the voltage across the NTC thermistor in a series circuit. The device has quick response, good repeatability, low cost, and is also easy to fabricate. Within the concentration range of 0 to 2.5% by volume, the hydrogen sensitivity obtained is up to −0.52 V/%.

## 2. Operation Principle

Figure 1a,b show the schematic and real photograph of the proposed sensor based on an NTC thermistor with Pt-loaded WO_3_/SiO_2_ coating. The thermistor chip and its two sintered electrodes are packaged in ethoxyline resin with two dumet wires left outside. The NTC thermistor (bought from Shenzhen Minchuang Electronics Co. Ltd., MF52, Guangdong, China) we utilized in our experiments has a small size (less than 2 mm), a resistance of 10 KΩ at the temperature of 25 °C, and is characterized by excellent precision and a rapid response. By using a dipping technique, a Pt-loaded WO_3_/SiO_2_ layer with a thickness of ~30 μm was applied on the surface of an NTC thermistor due to the strong Van der Waals force. The Pt-loaded WO_3_/SiO_2_ catalyst was prepared by the sol-gel method [29]. The morphology of the coating was characterized using a field emission scan electronic microscope (FE-SEM SU8010, Hitachi, Tokyo, Japan), as shown in Figure 1c. Obviously, the Pt-loaded WO_3_/SiO_2_ is primarily composed of aggregated particles of similar size and the aggregation is loose and porous, allowing hydrogen molecules to diffuse and increasing the surface area of the contact. Additionally, the elemental analysis was also performed using energy dispersive spectrometry (EDS) with an x-ray detector attached to the FE-SEM instrument. The EDS pattern is shown in Figure 1d, and it can be concluded that the molar ratio of Pt: WO_3_ is approximately 1:5 in order to achieve good sensitivity and stability for hydrogen sensing [29,30].

Figure 2 shows the experimental setup to measure hydrogen concentration. It contains a NTC thermistor (MF52-10 KJ) with Pt-loaded WO_3_/SiO_2_ coating, a fixed 10 kΩ resistor and a 5 V DC power supply. The NTC thermistor and the fixed resistor are connected in a series. The NTC thermistor with Pt-loaded WO_3_/SiO_2_ coating is placed in a gas chamber. A gas flow controller (Sevenstar, EF-03, Beijing, China) with two channels is used in the experimental setup. The first channel has a range of 0–5 standard-state cubic centimeters per minute (SCCM), and the second channel has a range of 0–500 SCCM. Hydrogen from a hydrogen generator flows into the gas chamber through the first channel and air from an air pump flows into the gas chamber through the second channel. The flow controller can precisely control the hydrogen concentration in volume. When the concentration of hydrogen in the gas chamber reaches a stable level, the computer records the voltage across the sensor using a data acquisition unit. Every time the hydrogen concentration was changed in our experiments, the gas chamber was vacuumed with a vacuum pump.

According to the circuit shown in Figure 2, the voltage across the NTC thermistor coated with Pt-loaded WO_3_/SiO_2_ can be written as:(1)$$V_{s}(T)=\frac{V_{c}\times R_{s}(T)}{R_{s}(T)+R_{0}}$$
where *V_c_* is the voltage of DC power supply, *R_s_*(*T*) is the resistance of NTC thermistor, and *R*_0_ is the resistance of fixed resistor. The resistance of NTC thermistor with respect to temperature can be described as [28]:(2)$$R_{S}(T)=10\exp[B(\frac{1}{T+273.15}-\frac{1}{298.15})]$$

Here, *B* is the negative temperature coefficient. The voltage across NTC thermistor will change if the temperature changes. From Equations (1) and (2), the temperature derivative of the voltage across the NTC thermistor can be obtained as:(3)$$\frac{dV_{S}(T)}{dT}=\frac{-50R_{0}\times \frac{B}{{\left(T+273.15\right)}^{2}}\times \exp[B(\frac{1}{T+273.15}-\frac{1}{298.15})]}{{\left(R_{0}+10\exp[B(\frac{1}{T+273.15}-\frac{1}{298.15})]\right)}^{2}}$$

According to Equation (3), the temperature sensitivity of the NTC thermistor is the highest, when the resistance of the fixed resistor is equal to the resistance of the NTC thermistor. Therefore, a fixed 10 kΩ resistor is used in the series circuit. The temperature coefficient of the thermistor is negative. As a result, as the temperature rises, the voltage of the NTC thermistor decreases. Temperature can be measured by monitoring the voltage across the NTC thermistor. Therefore, the NTC thermistor can function as a temperature sensor.

When the NTC thermistor with Pt-loaded WO_3_/SiO_2_ coating is exposed to the mixture of air and H_2_, owing to the existence of Pt, the exothermic reactions occur in catalytic layers. These reactions can be written as [3]:(4)$${\mathrm{WO}}_{3}+{\mathrm{xH}}_{2}\overset{\mathrm{Pt}}{\to }{\mathrm{WO}}_{3-x}+{\mathrm{xH}}_{2}O$$
(5)$${\mathrm{WO}}_{3-x}+\frac{x}{2}O_{2}\overset{\mathrm{Pt}}{\to }{\mathrm{WO}}_{3}$$

Under constant hydrogen concentration, these reactions continuously generate heat and raise the temperature of the NTC thermistor until a thermal equilibrium is reached. As a result, the concentration of hydrogen can be determined by monitoring the change of the voltage across the NTC thermistor caused by the temperature changes.

## 3. Experimental Results and Discussions

Firstly, the temperature response of the NTC thermistor without Pt-loaded WO_3_/SiO_2_ coating was tested. The thermistor was placed in the slot of a copper block which is pasted onto a thermoelectric cooler (with an accuracy of ±0.1 °C). The temperature varied from 0 to 45 °C in a step of 5 °C. The voltage across the NTC thermistor was converted into a digital signal by a data acquisition unit and recorded by a computer. Figure 3a shows the voltage across the NTC thermistor under various temperatures. Obviously, the voltage of the NTC thermistor decreases with the increase of environmental temperature, which can be explained by Equation (3). Furthermore, Figure 3b gives the final voltage across the NTC thermistor versus the temperature. A linear fitting shows that the sensor exhibits a temperature sensitivity of −0.056 V/°C and a good linear response (R^2^ = 0.9975). It is also noticed that the voltage across the NTC thermistor is constant at room temperature.

Next, the response of the NTC thermistor to hydrogen was measured. The NTC thermistor with Pt-loaded WO_3_/SiO_2_ coating was placed in the gas chamber. Pure hydrogen from a hydrogen generator and air from an air pump flowed into the gas chamber through the two channels of the flow controller. The hydrogen generator was linked to the channel with a range of 0–5 SCCM and the air pump was connected to the other channel with a wider range of 0–500 SCCM. The concentration of hydrogen was adjusted by the flow controller. 

Figure 4a depicts the voltage across the NTC thermistor with a Pt-loaded WO_3_/SiO_2_ coating at various hydrogen concentrations of 0%, 0.5%, 1%, 1.5%, 2%, and 2.5% by volume in the gas chamber at a room temperature of ~25℃. Obviously, the voltage across the NTC thermistor decreases as the hydrogen concentration increases, which can be explained by the increase of temperature caused by the exothermic reactions as described in Equations (4) and (5). More heat is released as hydrogen concentration increases. Figure 4b presents the final voltage versus hydrogen concentration within the range of 0–2.5% by volume. It can be seen that the change of voltage across the NTC thermistor is nonlinear to the hydrogen concentration. A polynomial fitting shows that the NTC thermistor has approximate quadratic function sensitivity with an Adj. R^2^ of 0.9982. The sensitivity increases with the increase of hydrogen concentration and reaches up to −0.52 V/% at the hydrogen concentration of 2.5% by volume. Considering safety, a higher hydrogen concentration than 2.5% by volume has not been tested in our experiments.

The repeatability of the sensor response to hydrogen was also tested. The sensor was quickly put into the gas chamber, where a constant hydrogen concentration of 2.0% was maintained, and then removed. Figure 5a shows the response of the sensor over three cycles. When the sensor is exposed to hydrogen, the voltage across the sensor dropped quickly and then stabilizes. When the sensor is removed from the gas chamber, however, the voltage gradually increased. The repeatability index *C_r_* can be expressed as:(6)$$C_{r}=\frac{SD_{r}}{C_{range}}\times 100\%$$
where $SD_{r}$ is the standard deviation of the voltage across the NTC when it reaches stability while the sensor is exposed to hydrogen at each period, and $C_{range}$ is the voltage difference at two hydrogen concentration stages (i.e., 0 and 2.0%). Based on this definition, the variation in the repeatability of the sensor is calculated with maximum $C_{r-\max}$ values of 0.32%, which indicates that the sensor has good repeatability and stability. Figure 5b presents the detailed response of the sensor to hydrogen, which shows the sensor has a fast response of ~2 min and a relatively long recovery of ~17.3 min by calculating the response signal from 10% $C_{r}$ to 90% $C_{range}$. The fast response is attributed to the Pt-loaded WO_3_/SiO_2_ particles of small size and the aggregation with porous, which causes a large surface and easy penetration of hydrogen molecules. The relatively long recovery time may be related to the package of the thermistor chip, the shell of WO_3_/SiO_2,_ and the environmental temperature. Directly coating the Pt-loaded WO_3_/SiO_2_ on the thermistor chip but not on the shell would be effective in reducing the response and recovery time. 

Finally, the sensor’s response to humidity in the air to the sensor was evaluated. Figure 6 shows the sensor final voltage versus hydrogen concentration in dry and wet air. In the experiments, three types of air with different relative humidity at a room temperature of ~25 °C were used. The dry air was obtained by flowing the indoor air into a drying tube filled with Na_2_SO_4_, and the wet air with a relative humidity of 30% was obtained directly from the indoor air, while the wet air with a relative humidity of 80% was obtained by flowing the indoor air through a conical flask filled with distilled water. It can be seen that the hydrogen sensor’s responses of the hydrogen sensor are similar for these three types of air with varying relative humidity. Therefore, air humidity has little effect on the sensor’s performance.

## 4. Conclusions

A hydrogen sensor based on an NTC thermistor with Pt-loaded WO_3_/SiO_2_ coating is presented and demonstrated. When the sensor is incorporated into a series circuit and exposed to the mixture of air and H_2_, the exothermic reaction increases the temperature of the NTC thermistor while decreasing the voltage across the NTC thermistor. We can precisely measure the hydrogen concentration by monitoring the voltage across the NTC thermistor in the series circuit.

The experimental results show that the sensitivity of the proposed sensors increases with hydrogen concentration, reaching up to −0.52 V/% at a hydrogen concentration of 2.5% by volume. A higher sensitivity can be expected when using an NTC thermistor with a larger temperature coefficient B. To pursue a small limit of hydrogen detection, a voltage data acquisition unit with a high resolution is required, for example, a voltage resolution of 1 mV corresponds to a hydrogen concentration variation of 20 ppm at 2.5% by volume. The ability to distinguish is very high, and it is superior to or comparable with other sensors [14,15,23,24]. Furthermore, our sensor has good repeatability and stability with a maximum variation value of 0.32% as well as immunity resistance, a small size (~2 mm in dimension), and a low fabrication cost.

It is noted that to ensure the long-term stable operation of the device, drastic vibration should be avoided, although hydrogen-sensitive materials of Pt-loaded WO_3_/SiO_2_ are coated on the surface of the device by a strong van der Waals force. Our future work will also include the cross-interference testing of other gases, such as CO, NO and CH_4_.

## Figures and Tables

**Figure 1 micromachines-13-02219-f001:**
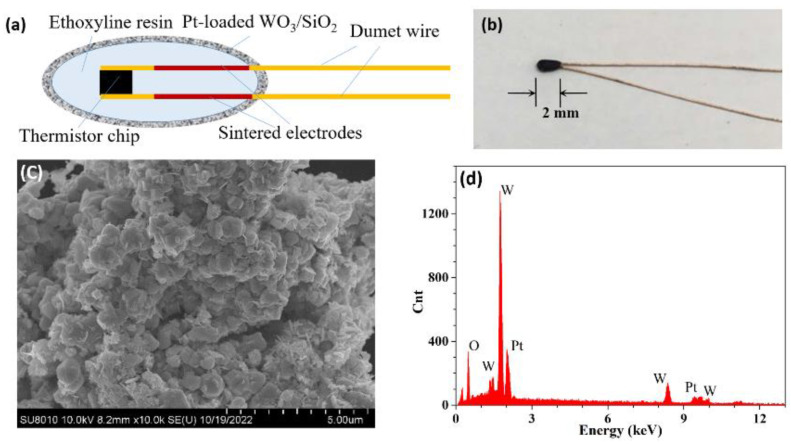
Characters of the proposed sensor. (**a**) Schematic graph and (**b**) photo of the NTC thermistor with Pt-loaded WO_3_/SiO_2_ coating. (**c**) Morphology and (**d**) EDS pattern of Pt-loaded WO_3_/SiO_2_ coating.

**Figure 2 micromachines-13-02219-f002:**
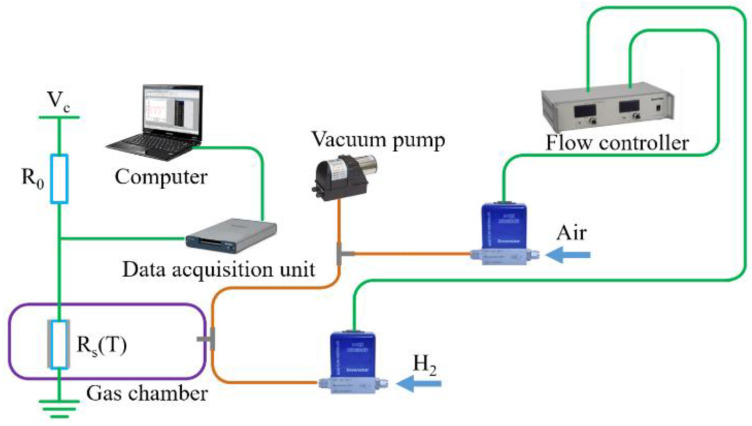
Experimental setup for hydrogen sensing. The sensor is installed in the gas chamber, where the hydrogen concentration is adjusted by controlling the air-to H_2_ flow ratio, and the voltage across the sensor is recorded by computer through the data acquisition unit.

**Figure 3 micromachines-13-02219-f003:**
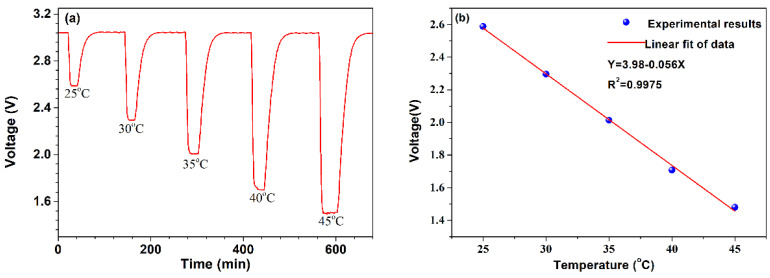
NTC thermistor response to temperature. (**a**) The voltage across the NTC thermistor within the temperature range of 25 °C–45 °C with a step of 5 °C. (**b**) Relationship between the voltage across the NTC thermistor and its temperature.

**Figure 4 micromachines-13-02219-f004:**
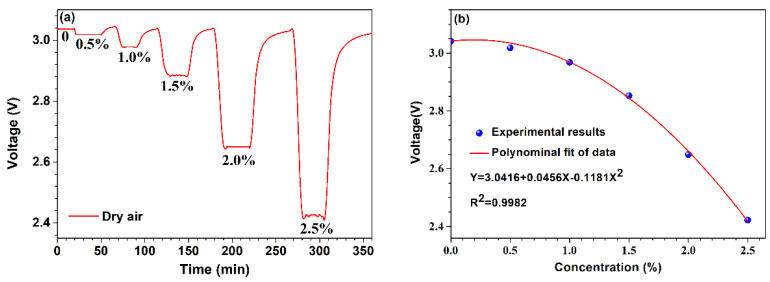
Sensor responses to hydrogen at room temperature of ~25 °C. (**a**) The voltage across the sensor within the hydrogen concentration ranges from 0 to 2.5% by volume. (**b**) The relationship between the voltage across the sensor and the hydrogen concentration.

**Figure 5 micromachines-13-02219-f005:**
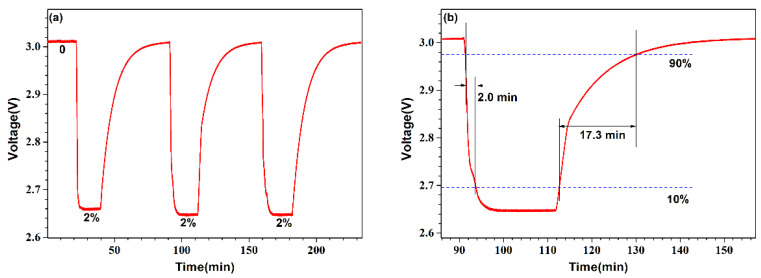
Time response of the sensor at room temperature of ~25 °C (**a**) upon consecutive cycles from a pure air without hydrogen to a mixture of 2% hydrogen and (**b**) the detailed response for a single cycle.

**Figure 6 micromachines-13-02219-f006:**
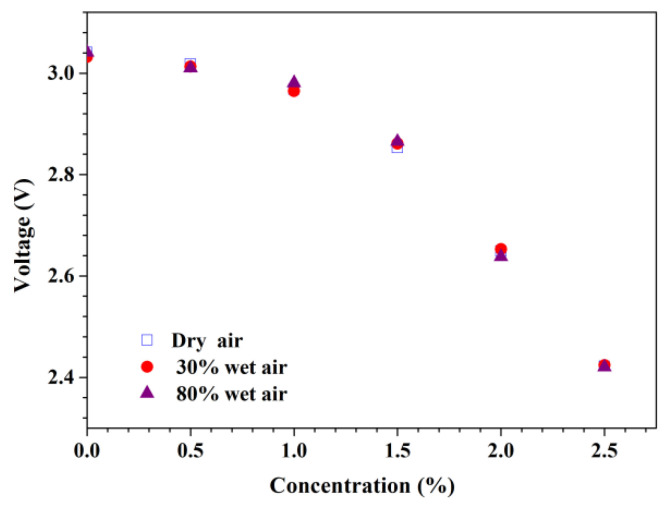
The response of the sensor to a mixture of hydrogen with different relative humidity at a room temperature of ~25 °C.

## Data Availability

The data that support the findings of this study are available from the corresponding author upon reasonable request.

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
