# Peer review of "Hydrogen Sensor Based on NTC Thermistor with Pt-Loaded WO3/SiO2 Coating"

_micromachines, 2022, doi:10.3390/mi13122219_

Round 1

Reviewer 1 Report

The authors report hydrogen sensor based on Pt-loaded WO3 SiO2 coating.  Overall results of the manuscript look good but the author should provide additional data to improve the paper. 

1.  The author should present long term stability of the sensors. 

2. Recent advancement of hydrogen sensing materials should be included in the introduction part. Flexible hydrogen sensors based on 2D materials decorated with noble metal nanoparticles have been reported.

e.g. Sens. Actuators B, 349 (2021) 130696.

Reviewer 2 Report

The authors present the experimental realization and characterization of a Pt-loaded WO3/SiO2 coating on an NTC thermistor for hydrogen sensing. The topic is interesting and very relevant. However, before the manuscript can be published, some improvements are necessary in my opinion. These are:

- you propose a sensor for hydrogen, which is relevant to detect mainly around its lower explosion limit (LEL) of 4%. However, you tested your sensor only to a max. concentration of 2.5%. Why not higher (say up to 10%)? Please insert a comment on this.

- the resolution of several figures is very poor. This should be improved.

- the figure captions are kept very short. They should explain in detail what the figures show, and must be understandable without knowing the main text.

- I'm missing a "materials" section. How did you prepare your sensors? Where did you buy the chemicals? etc. Please include this.

- the conclusions are also kept quite short. Here you should go into greater detail what your sensor can reach and how this compares to the state-of-the art.

Round 2

Reviewer 2 Report

Thanks to the authors for their thorough revision of the manuscript which I would now consider as acceptable for publication.

Author Response

Thank you. The manuscript has been thoroughly revised.